# Waterproof and ultraflexible organic photovoltaics with improved interface adhesion

Sixing Xiong[1], Kenjiro Fukuda[1,2] ✉, Kyohei Nakano[1], Shinyoung Lee[1], Yutaro Sumi[3], Masahito Takakuwa[3,4], Daishi Inoue[1], Daisuke Hashizume[1], Baocai Du[1,3], Tomoyuki Yokota[3,4], Yinhua Zhou[5], Keisuke Tajima[1] & Takao Someya[1,2,3] ✉

Ultraflexible organic photovoltaics have emerged as a potential power source for wearable electronics owing to their stretchability and lightweight nature. However, waterproofing ultraflexible organic photovoltaics without compromising mechanical flexibility and conformability remains challenging. Here, we demonstrate waterproof and ultraflexible organic photovoltaics through the in-situ growth of a hole-transporting layer to strengthen interface adhesion between the active layer and anode. Specifically, a silver electrode is deposited directly on top of the active layers, followed by thermal annealing treatment. Compared with conventional sequentially-deposited hole-transporting layers, the in-situ grown hole-transporting layer exhibits higher thermodynamic adhesion between the active layers, resulting in better waterproofness. The fabricated 3 μm-thick organic photovoltaics retain 89% and 96% of their pristine performance after immersion in water for 4 h and 300 stretching/releasing cycles at 30% strain under water, respectively. Moreover, the ultraflexible devices withstand a machine-washing test with such a thin encapsulation layer, which has never been reported. Finally, we demonstrate the universality of the strategy for achieving waterproof solar cells.

Water is a significant factor in the degradation of flexible electronics[1,2]. Water condensation can occur on the device surfaces because of rain or high humidity outdoors, and everyday indoor activities like washing hands can result in water attached to cloth and skin[3,4]. Ultraflexible organic photovoltaics (OPVs) are light and can be deformed by using a substrate with a thickness of several μm or less, making them compatible with wearable and skin electronics[5,6]. Hence, waterproofness becomes a critical parameter when utilizing ultraflexible OPVs as energy-harvesting solutions for wearable electronics that are integrated into daily life[7,8].

Typically, enhancing the waterproofness of OPVs involves embedding them in a thick or rigid encapsulation layer[9]. For example, flexible OPVs with double-sided, 500 μm-thick encapsulating layers maintained approximately 95% of their initial power conversion efficiency (PCE, ~7.9%) after being immersed in water for 120 min[10]. Another textile-based OPV (PCE ~ 7.2%) encapsulated on both sides by a tens of microns thick composite layer of atomic-layer-deposited aluminum oxide and $SiO_2$–polymer barriers demonstrated a 2% loss in efficiency after being washed for 20 cycles (10 min per cycle) under stirring at 200 rpm using a magnetic stirrer[11].

[1]RIKEN Center for Emergent Matter Science (CEMS), Wako 351-0198 Saitama, Japan. [2]Thin-Film Device Laboratory, RIKEN, 2-1 Hirosawa, Wako 351-0198 Saitama, Japan. [3]Department of Electrical Engineering and Information Systems, The University of Tokyo, 113-8656 Tokyo, Japan. [4]Institute of Engineering Innovation, The University of Tokyo, 113-8656 Tokyo, Japan. [5]Wuhan National Laboratory for Optoelectronics, Huazhong University of Science and Technology, Wuhan, China. ✉e-mail: kenjiro.fukuda@riken.jp; takao.someya@riken.jp

However, achieving improved waterproofness without compromising mechanical flexibility and conformability remains a challenge. Covering with thick or rigid encapsulation layers to enhance waterproofness inevitably impair conformability, stretchability, and even output performance of wearable electronics[12]. On the other hand, the water sensitivity of conventional hole-transporting layer (HTL) materials would induce performance degradation of OPVs in the presence of water[13,14]. Additionally, deformation typically increases moisture transmission rate of elastic or flexible encapsulating materials[15], thereby accelerating the deterioration of the devices in the presence of water or moisture.

Here, we present an ultraflexible OPV featuring an in-situ growth of AgO$_x$ HTL, ensuring significant waterproofness even under mechanical deformation by strengthening the electrode and active layer interface adhesion. The in-situ growth procedure entails the direct deposition of Ag on active layers, followed by annealing in air at 85 °C for 24 h. Consequently, our 3 μm-thick OPVs exhibit significantly improved waterproofness compared to those using conventional HTL materials, retaining 89% and 96% of the initial efficiency after 240 min of immersion and 300 stretching/releasing cycles at 30% strain underwater, respectively. These features enable our ultraflexible OPVs to withstand machine-washing processes. With a champion efficiency of 14.3% under one sun illumination, our developed OPVs outperform existing waterproof OPVs (Supplementary Table 1)[10,11,16,17]. Furthermore, our approach demonstrates universality in achieving waterproof OPVs with different active layers. The strategy significantly improves the waterproofness from the fundamental structure, enhancing waterproofness of OPVs without compromising mechanical durability and conformability. Our developed approach opens possibilities for wearable energy systems and propels the progress of waterproof electronics.

## Results

### Device with in-situ growth of HTL

The waterproof and ultraflexible OPVs were realized by growing of AgO$_x$ HTL in situ with a structure of transparent polyimide (tPI)/ITO/PEI-Zn/PM6:Y6/AgO$_x$/Ag/Parylene (Fig. 1a). The Ag electrode was directly deposited onto an active layer of PM6:Y6 (Supplementary Fig. 1), followed by annealing at 85 °C in air. The devices were encapsulated with a 1 μm-thick parylene layer after annealing treatment. Before annealing, the fabricated OPVs showed poor efficiency−open-circuit voltage ($V_{OC}$) of 0.05 V, short-circuit current density ($J_{SC}$) of 15.8 mA cm$^{-2}$, fill factor (FF) of 0.27, PCE of 0.2% (Fig. 1b), and an average PCE of 0.2% (Table 1). After 12 h of annealing, the device performance improved−$V_{OC}$ = 0.77 V, $J_{SC}$ = 25.8 mA cm$^{-2}$, FF = 0.66, PCE = 13.1%, and an average PCE of 12.5%. Extending the annealing to 24 h further improved the performance−$V_{OC}$ = 0.77 V, $J_{SC}$ = 26.5 mA cm$^{-2}$, FF = 0.71, PCE = 14.3%, and an average PCE of 13.6%. Supplementary Fig. 2 displays the PCE distribution based on 30 separate solar cell devices after 24 h of annealing at 85 °C.

The external quantum efficiency (EQE) spectrum showed significant enhancement after annealing (Fig. 1c). Devices with 24 h annealing did not exhibit hysteresis for the forward and reverse scans (Supplementary Fig. 3), indicating efficient extraction and collection of charge carriers. Free-standing devices were delaminated from the supporting glass and seamlessly attached to human skin (Supplementary Fig. 4). The devices demonstrated excellent storage stability, preserving 92% of the pristine PCE after 1369 h of air storage in

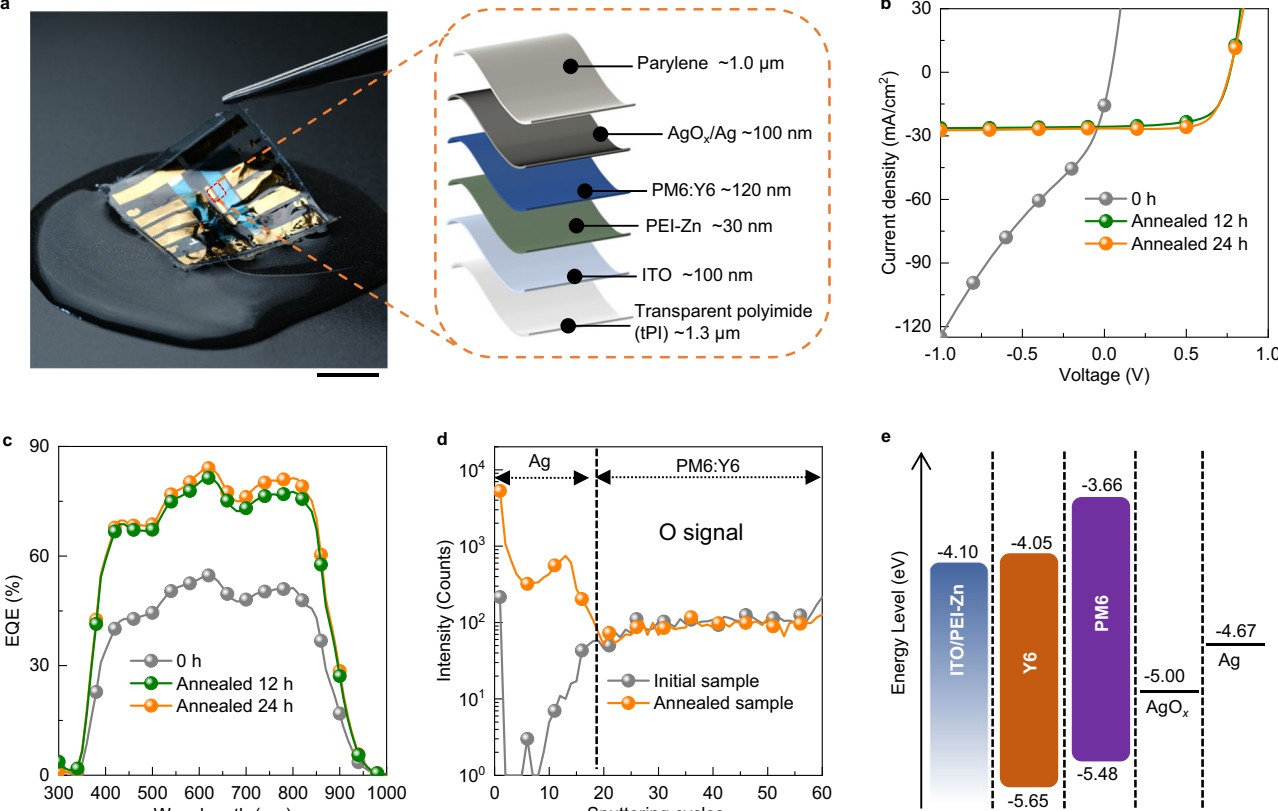

**Fig. 1 | Structure, characteristics, and mechanism analysis of waterproof OPVs.** **a** Photograph of the ultraflexible and waterproof OPVs floating on water. Schematic device structure of the waterproof OPVs. Scale bar: 1 cm. **b** J–V characteristics of devices with 0 h, 12 h, and 24 h annealing treatment at 85 °C in air. **c** The evolution of EQE of the OPVs under different annealing times at 85 °C. **d** Depth profiles of O element determined by D-SIMS for initial and annealed samples. **e**, Energy level diagram of annealed samples.

**Table 1 | Photovoltaic parameters of the OPVs based on different annealing times**

| Annealing time | $J_{SC}$ (mA cm$^{-2}$) | $V_{OC}$ (V) | FF | PCE (%) |
|---|---|---|---|---|
| 0 h | 15.8 (15.8 ± 1.9) | 0.05 (0.05± 0.01) | 0.27 (0.27 ± 0.01) | 0.2 (0.21 ± 0.08) |
| 12 h | 25.8 (25.4 ± 0.6) | 0.77 (0.77 ± 0.01) | 0.66 (0.64 ± 0.02) | 13.1 (12.51 ± 0.47) |
| 24 h | 26.5 (26.2 ± 0.5) | 0.77 (0.76 ± 0.02) | 0.71 (0.68 ± 0.02) | 14.3 (13.6 ± 0.4) |

Parentheses show the average value along with the standard deviation.
These are statistical values of average and standard deviation obtained from 30 samples.

darkness at room temperature (20–25 °C) and 93% of their original PCE after 1371 h of aging at 85 °C in air (Supplementary Fig. 5). These results are comparable to recently reported organic photovoltaics with HTLs (Supplementary Table 2)[10,18–25]. Additionally, the devices maintained 87.7% of their initial efficiency under AM 1.5 G illumination with a filter cutting off light below 400 nm in a N$_2$-filled glovebox for 334 h (Supplementary Fig. 6), surpassing recently reported ultrathin OPVs (Supplementary Table 3)[24,26–30]. The temperature and annealing time optimizations of in-situ growth of AgO$_x$ HTL are shown in Supplementary Fig. 7. The devices annealed for 24 h at 85 °C demonstrated the best average performance (Supplementary Table 4), highlighting the crucial role of the appropriate annealing in the formation of an efficient AgO$_x$ HTL. Supplementary Fig. 8 illustrates the evolutions of different annealing times of photocurrent density ($J_{ph}$)–effective voltage ($V_{eff}$) curves and light intensity-dependent $J_{SC}$. These results suggest that devices annealed for 24 h at 85 °C experienced a significant improvement in the collection and extraction of charge carriers, as well as the suppression of bimolecular recombination. However, the excessive annealing can cause performance deterioration[24,31]. Consequently, the prolonged heating could not yield further enhancements in device performance. Additionally, the proposed thermal strategy for in-situ growth of AgO$_x$ HTL exhibited much better performance compared with natural oxidation treatment[32] reported in a previous work (Supplementary Fig. 9 and Supplementary Table 5), indicating that moderate thermal stress could promote the growth of AgO$_x$.

Dynamic secondary ion mass spectrometry (D-SIMS) measurement was conducted to analyze the in-situ growth of AgO$_x$ HTL. Fig. 1d displays the depth profiles of O signal intensities of glass/ITO/PM6:Y6/ Ag before and after annealing. The Ag and PM6:Y6 layer regions are identified based on F element distribution (Supplementary Fig. 10). After 24 h of annealing at 85 °C, the O signal intensities in Ag layer improved significantly. A peak at the Ag and PM6:Y6 interface suggests the oxygen diffused from the interface to form the AgO$_x$ HTL layer[32,33]. In addition, annealing the devices with an Ag electrode at 85 °C for 24 h in an N$_2$-filled glovebox did not show any improvement in PCE (Supplementary Fig. 11), indicating that annealing treatment in air is necessary for AgO$_x$ HTL in-situ growth.

Subsequently, we checked the work function of the Ag electrode before and after annealing. Notably, the work function of top surface of the annealed Ag electrode is almost the same as that of a fresh Ag electrode, and the conductivity remains unchanged (Supplementary Fig. 12). These indicate that the 24 h annealing does not cause significant oxidation on the Ag surface, thereby not affecting the performance. Here, we applied a removable electrode[34] to assess the work function of the bottom surface of the Ag electrode in contact with the active layer (Supplementary Fig. 13a). The annealed removable Ag electrode exhibited a work function of approximately 5.0 eV (Supplementary Fig. 13b), similar to the reported value[35]. Notably, silver oxidizes faster in the Ag-organic interface during annealing compared with the top surface of the Ag electrode because oxygen diffusion across grain boundaries is quicker than lattice diffusion[36]. The silver electrode generally grows in the Volmer–Weber mode ('island' mode) on an active layer[37], leading to a rougher surface and higher grain-boundary density in the Ag–organic interface than at the top surface of

Ag electrode[38]. The energy-level diagram of the optimized devices is shown in Fig. 1e.

## Stability in water
We tested the waterproofness of the free-standing OPVs by immersing them in water (Fig. 2a). The OPV with an AgO$_x$/Ag electrode exhibited only a 11% decrease in PCE after 4 h of water immersion (Fig. 2b). Additionally, we explored the influence of water temperature on the waterproofness. All devices based on AgO$_x$ HTL demonstrated significant waterproofness under 0 °C, 20 °C, and 40 °C water immersion (Supplementary Fig. 14). The results indicate that the waterproof performance remains largely unaffected by water temperatures in the range of 0 °C–40 °C. For comparison, devices having a structure of tPI/ ITO/PEI-Zn/PM6:Y6/MoO$_x$/Ag/Parylene were also fabricated (Supplementary Fig. 15). However, the device with MoO$_x$/Ag experienced a complete performance drop after only 2 h of water immersion. Then, we examined the electrode conductivity of the waterproof devices before and after water immersion. Both AgO$_x$/Ag and MoO$_x$/Ag electrodes maintained a stable conductivity after water immersion (Supplementary Fig. 16a, b), implying that the electrode conductivity does not impact the water stability of the devices. To delve deeper, impedance tests were conducted, and the impedance spectra of the device before and after water immersion are shown in Supplementary Fig. 16c, d. The transport resistance[39] (intersecting point of the curve and the lateral axis at low frequencies) of the device based on AgO$_x$/Ag slightly increased after water immersion, whereas that of the device with MoO$_x$/Ag sharply increased only after 1 h immersion. The increased resistance indicates an increase in charge recombination after water immersion[39–41]. Because both devices share the same structure except for the HTL layer, the observed difference can be attributed to the charge transportation within the HTL layer. The OPV device with the AgO$_x$/Ag electrode also works when fully immersed in water under light illumination (Supplementary Fig. 17). Additionally, we compared the long-term operation of free-standing OPVs in water with different HTLs (Supplementary Fig. 18). Fig. 2c shows the $J_{SC}$ of the device with AgO$_x$/Ag and MoO$_x$/Ag electrode under light illumination in water. The current of the developed OPV with AgO$_x$/Ag electrode remained above 91% of its initial value after operation in water for 64 min. In contrast, the device with MoO$_x$/Ag electrode exhibited a rapid deterioration, with a loss of 35% after operation in water for 24 min.

Subsequently, we subjected the devices to a cyclic compression test to assess their mechanical durability underwater. A photograph of free-standing devices under deformation stress in water is shown in Fig. 2d. The cyclic compression test under water involves a combination of stresses, including water exposure and deformation pressure, which impose stringent demands on device stability. The testing equipment for cyclic compression test is depicted in Supplementary Fig. 19. The OPV with AgO$_x$/Ag electrode retained 96% of its initial value under 30% strain after 300 cycles of stretching–compressing in water (Fig. 2e). In contrast, the device with MoO$_x$/Ag electrode saw its PCE plummet to 0% after just 100 stretching–compressing cycles. The sharp decline indicates that the mechanical stress underwater severely compromises the waterproofness of devices with MoO$_x$/Ag electrodes. The results emphasize that OPVs with AgO$_x$/Ag electrodes maintain

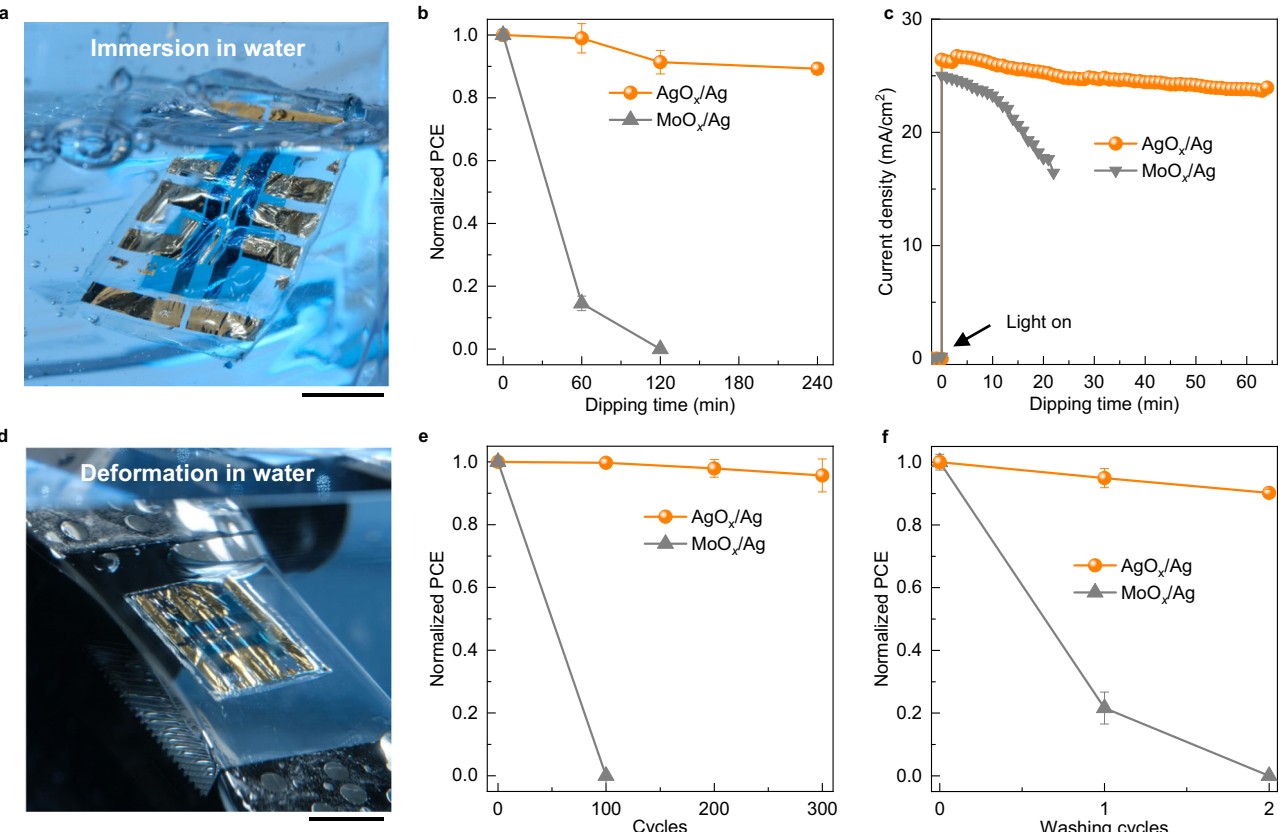

**Fig. 2 | Stability of free-standing devices in water. a** Photograph of the free-standing devices being immersed in water. Scale bar: 1 cm. **b** The evolution of PCE for PM6:Y6 devices based on AgO$_x$/Ag and MoO$_x$/Ag electrodes with different immersion times in water. **c**, The evolution of $J_{SC}$ for PM6:Y6 devices based on AgO$_x$/Ag and MoO$_x$/Ag electrodes as a function of dipping time in water under light illumination. **d** Photograph of the free-standing devices under deformation stress in water. Scale bar: 1 cm. **e** The evolution of PCE for PM6:Y6 devices based on AgO$_x$/Ag and MoO$_x$/Ag electrodes under cyclic compression test in water. **f** The evolution of PCE for PM6:Y6 devices based on AgO$_x$/Ag and MoO$_x$/Ag electrodes under washing process by washing machine.

stability even when subjected to mechanical deformation underwater, highlighting their suitability for wearable electronics.

Furthermore, given the excellent water stability of our devices even under deformation, we conducted the machine-washing test. Fig. 2f illustrates the machine-washing test results for free-standing OPVs without detergent. OPVs with AgO$_x$/Ag electrodes remained 90% of their original PCE after two washing cycles in a washing machine. However, devices with MoO$_x$/Ag electrodes experienced a complete PCE drop after two washing cycles. Notably, the concept of ultra-flexible OPVs enduring machine-washing has not been reported in previous literature. These outcomes underscore that the developed OPVs meet the rigorous requirements of machine-washing tests. Furthermore, these findings suggest that OPVs with AgO$_x$/Ag electrode can maintain continuous operation underwater, thereby enabling long-running power source for underwater autonomous systems and sensors[42]. The substitution of MoO$_x$ with AgO$_x$, achieved through a novel thermal annealing strategy, advanced the waterproofness of OPVs. With HTL optimization, the water stability of OPVs improved from the aspect of fundamental structure perspective. We believe that the combination of AgO$_x$ HTL and thermal annealing presents an alternative potential strategy for designing waterproof electronics.

## Interface adhesion strength

To identify the primary source of degradation in devices with MoO$_x$/Ag electrodes exposed to water, we fabricated devices with the structure of glass/ITO/PEI-Zn/PM6:Y6/MoO$_x$/Ag and immersed them in water. As shown in Fig. 3a, the top electrode is completely removed from the active layer after 1 h of immersion. To investigate further, we

separately redeposited the Ag and MoO$_x$/Ag electrodes onto the active layer. The device with the redeposited Ag electrode exhibited a poor efficiency of 0.6% (Fig. 3b), whereas the device with the redeposited MoO$_x$/Ag electrode exhibited a PCE comparable to the initial device (Fig. 3c and Table 2). This indicates that delamination occurred at the interface between MoO$_x$ and the active layer during water immersion, leading to PCE deterioration.

Fig. 3d presents the $J$–$V$ characteristic curves of the device with structure of glass/ITO/PEI-Zn/PM6:Y6/AgO$_x$/Ag before and after immersion in water for 1 h. The device exhibited a loss in performance of only 2.2% after immersion (Table 2). In addition, we conducted a waterproofness test on naturally oxidized devices (Supplementary Fig. 20). The naturally oxidized devices demonstrated significant PCE deterioration (10.3% loss) after 1 h of water immersion (Supplementary Table 6). This could be attributed to the improved bonding strength between the active layer and electrode due to thermal annealing, compared to natural oxidation treatment (Supplementary Fig. 21), thereby highlighting the importance and advancement of the thermal annealing strategy. A thermally stable interface between the electron-transporting layer (ETL) and active layer is essential for the success of this strategy. Fortunately, our previous work successfully established a thermally stable interface using the PEI-Zn ETL[24]. Additionally, we fabricated devices with the structure of glass/ITO/PEI-Zn/PM6:Y6/PEDOT:PSS/Ag. Similar to devices with MoO$_x$/Ag electrodes, water exposure resulted in delamination between the top electrode and active layer (Supplementary Fig. 22). These results clearly demonstrate the superior reliability of the AgO$_x$/Ag structure compared to MoO$_x$/Ag or PEDOT:PSS/Ag when exposed to water.

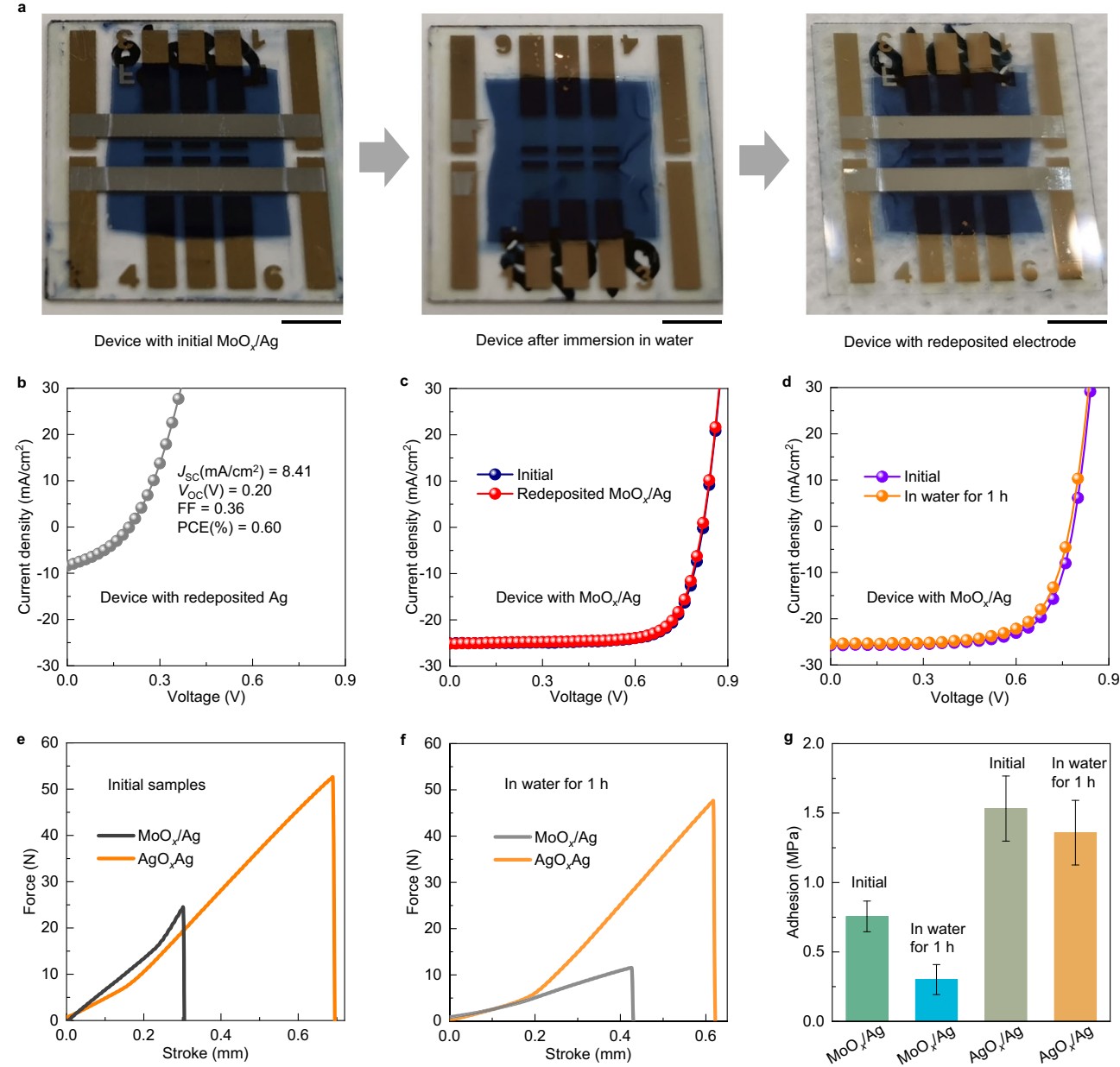

**Fig. 3 | Mechanism analysis for enhancing waterproofness. a** Photograph of the device with initial $MoO_x/Ag$ (left), after immersion in water for 1 h (middle), and redeposited metal electrode (right). Scale bar: 5 mm. **b** $J–V$ characteristics of PM6:Y6 devices after 1 h of immersion with redeposited Ag. **c** $J–V$ characteristics of PM6:Y6 devices with initial and redeposited $MoO_x/Ag$. **d** $J–V$ characteristics of PM6:Y6 devices with $AgO_x/Ag$ electrode before and after immersing in water for 1 h. **e** Force stroke curves of initial samples with $AgO_x/Ag$ and $MoO_x/Ag$. **f** Force stroke curves of samples with $AgO_x/Ag$ and $MoO_x/Ag$ after being immersed in water for 1 h. **g** Comparison of adhesion for $AgO_x/Ag$ and $MoO_x/Ag$ electrodes on the PM6:Y6 active layer. The error bars in the plots of adhesion indicate standard deviations (based on three samples).

**Table 2 | Photovoltaic parameters of OPVs before and after immersion in water under 100 mW cm$^{-2}$ AM 1.5 G illumination**

| Anode | Situation | $J_{SC}$ (mA cm$^{-2}$) | $V_{OC}$ (V) | FF | PCE (%) |
|---|---|---|---|---|---|
| $MoO_x/Ag$ | Before immersion | 25.1 (24.9 ± 0.3) | 0.82 (0.82 ± 0.01) | 0.74 (0.73 ± 0.01) | 15.3 (14.8 ± 0.5) |
| | After immersion | / | / | / | / |
| | Re- deposited | 25.1 (24.6 ± 0.5) | 0.82 (0.82 ± 0.01) | 0.74 (0.72 ± 0.02) | 15.2 (14.3 ± 0.5) |
| $AgO_x/Ag$ | Before immersion | 25.8 (25.7 ± 0.3) | 0.79 (0.78 ± 0.01) | 0.69 (0.68 ± 0.01) | 14.0 (13.6 ± 0.3) |
| | After immersion | 25.6 (25.5 ± 0.2) | 0.77 (0.77 ± 0.01) | 0.68 (0.68 ± 0.02) | 13.4 (13.3 ± 0.4) |

The average efficiency as well as standard deviation is shown in parentheses.
These are statistical values of average and standard deviation for the obtained from six samples.

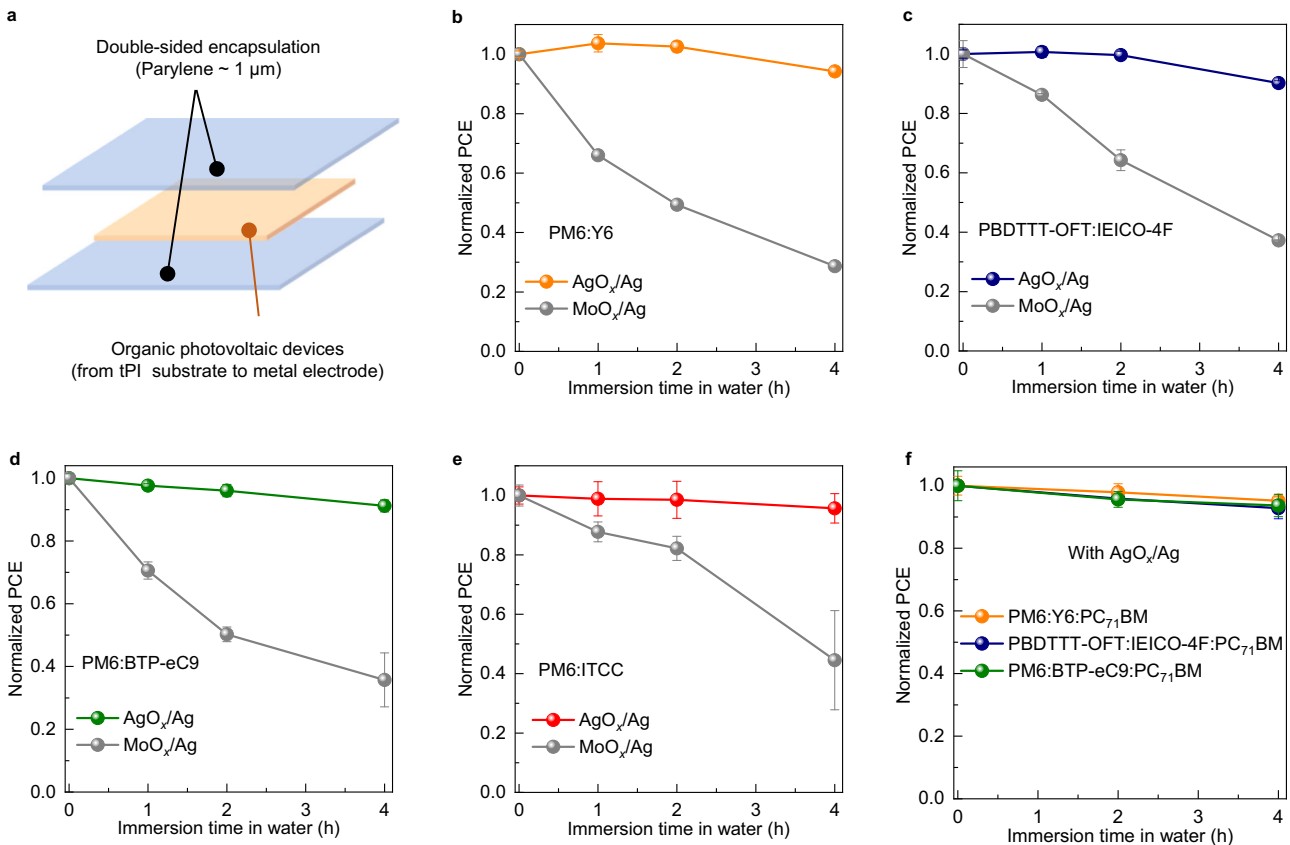

**Fig. 4 | Universality of developed structure in waterproofness improvement.**
**a** Schematic diagram of device with double-sided encapsulation layer. **b**–**f** The
evolution of PCE for double-side encapsulated devices based on different active
layers after being immersed in water. **b** Free-standing OPVs based on PM6:Y6 active
layer with AgO$_x$/Ag and MoO$_x$/Ag electrodes. **c** Free-standing OPVs based on
PBDTTT-OFT:IEICO-4F active layer with AgO$_x$/Ag and MoO$_x$/Ag electrodes. **d** Free-

standing OPVs based on PM6:BTP-eC9 with AgO$_x$/Ag and MoO$_x$/Ag electrodes.
**e** Free-standing OPVs based on PM6:ITCC active layer with AgO$_x$/Ag and MoO$_x$/Ag
electrodes. **f** Free-standing OPVs based on ternary active layer of PM6:Y6:PC$_{71}$BM,
PBDTTT-OFT:IEICO-4F:PC$_{71}$BM, and PM6:BTP-eC9:PC$_{71}$BM with AgO$_x$/Ag electro-
des. The error bars in the plots indicate standard deviations (based on six devices).

To quantitatively compare delamination forces between different
HTLs and active layer, a stretching machine was employed for a tensile
test (Supplementary Fig. 23). Cross-sectional scanning electron
microscopy (SEM) images revealed that the delamination occurred at
the interface between the active layer and HTLs during peeling-off test
(Supplementary Fig. 24), indicating that the tensile force can be used
to evaluate the strength between HTLs and the active layer. The tensile
force between AgO$_x$ and active layer was 51.0 N, whereas the force
between MoO$_x$ and active layer was 24.5 N (Fig. 3e). After water
immersion, the tensile force between AgO$_x$ and active layer remained
at 47.6 N, whereas the force between MoO$_x$ and active layer decreased
to 11.5 N (Fig. 3f). The slight decrease in the tensile force for AgO$_x$/Ag
after immersion can be attributed to the swelling of organic materials
in water. Importantly, even after 1 h of immersion, the tensile force
between AgO$_x$ and the active layer remained higher than that of MoO$_x$
without immersion.

Adhesion (σ) is defined as $\sigma = \frac{P}{\pi a^2}$, where $P$ is the critical force at
interfacial failure and $a$ is the radius of the stud[43]. Fig. 3g illustrates
that the adhesion between AgO$_x$ and the active layer demonstrates a
value of 1.53 MPa, which is twice that between MoO$_x$ and the active
layer (0.76 MPa). After 1 h of water immersion, the adhesion
between AgO$_x$ and the active layer remained relatively stable at
1.36 MPa, whereas that between MoO$_x$ and active layer decreased to
0.30 MPa. The interface adhesion strength between the active layer
and anodes (AgO$_x$/Ag and MoO$_x$/Ag) can be quantified using the
thermodynamic work of adhesion ($W_a$). $W_a$ represents the work
needed to separate two in-contact materials[44] and is related to the

surface energy ($\gamma$) of these materials. The $\gamma$ of a film can be char-
acterized by measuring the contact angles of liquid drops on the
film[45]. We measured the contact angles of deionized water and
glycerol drops on films (Supplementary Fig. 25), and we utilized
average surface energies of water and glycerol for accurate
calculations[46]. The dispersive surface energy ($\gamma^d$) and polar surface
energy ($\gamma^p$) of AgO$_x$ are 29.67 mJ/m$^2$ and 1.67 mJ/m$^2$, respectively,
resulting in a surface energy of 31.34 mJ/m$^2$. For MoO$_x$, the $\gamma^d$ and $\gamma^p$
are 3.41 mJ/m$^2$ and 80.21 mJ/m$^2$, respectively, yielding a surface
energy of 83.62 mJ/m$^2$ (Supplementary Table 7). Additionally, the
different polarities of AgO$_x$ ($\gamma^d > \gamma^p$) and MoO$_x$ ($\gamma^d < \gamma^p$) indicate
that the calculation of $W_a$ should be different (See Methods section
for detailed information). These differences in polarities and sur-
face energies result in varying $W_a$ values between the PM6:Y6 active
layer and AgO$_x$/Ag (44.08 mJ/m$^2$) and MoO$_x$/Ag (24.55 mJ/m$^2$)
anodes. A higher $W_a$ between the active layer and anode indicates a
stronger interface adhesion. Hence, the replacement of the anode
from MoO$_x$/Ag to AgO$_x$/Ag significantly improves the interface
adhesion. Moisture permeation from the interface could lead to
increased defect density[47] and delamination[48], resulting in effi-
ciency deterioration[49–51]. The strengthened interface plays a sig-
nificant role in preventing the delamination of the electrode and
active layer. Additionally, the water contact angle of MoO$_x$ is 6.3°,
whereas that of AgO$_x$ is 92.8°. A higher water contact angle indicates
better moisture resistance[52]. These findings underscore the super-
ior water stability of the AgO$_x$/Ag structure, making it the preferred
choice for water-exposed environments.

## Universality of the waterproof strategy

To improve the waterproofness of the proposed ultrathin freestanding OPVs[10], additional 1 μm-thick parylene encapsulation layer was added to the side of the tPI substrate (Fig. 4a). Following this, the double-sided encapsulated PM6:Y6 OPVs with $AgO_x$/Ag electrode maintained over 92% of their original PCE after 4 h of immersion. In contrast, devices with $MoO_x$/Ag electrodes exhibited a decline of their pristine PCE to 28% (Fig. 4b). It is noteworthy that while double-sided encapsulation indeed proves efficacious in enhancing the waterproofness of OPVs with $MoO_x$/Ag electrodes, it still falls short in waterproofing OPVs incorporating conventional HTLs. The encapsulation layer of parylene has a water vapor transmission rate (WVTR) of approximately 90 g/m² per day[18], inadequate for fully preventing the impact of water on the devices[53]. Moreover, we conducted the maximum power point tracking (MPPT) test with devices immersed in water. The device with $MoO_x$/Ag electrode diminished the performance to almost zero within 20 min. In contrast, the device with $AgO_x$/Ag electrode maintained 85.8% of its initial efficiency after operation in water for 70 min (Supplementary Fig. 26). These results underscore the exceptional waterproofness exhibited by our devised OPVs under operational conditions. In addition, the operational stability in water fluctuates considerably and decreases faster than that in air. The result may be attributed to the heat generated during maximum power point operation, causing the expansion of conductive tape connections, thereby weakening the waterproof capabilities of the encapsulation layer.

Subsequently, we examined the water stability of the OPVs with different active layers, including PBDTTT-OFT:IEICO-4F, PM6:BTP-eC9, and PM6:ITCC (Supplementary Fig. 1). Devices with $AgO_x$/Ag electrodes exhibited PCE values comparable to those with $MoO_x$/Ag electrodes (Supplementary Fig. 27a–f and Supplementary Table 8). As depicted in Fig. 4c–e, all $AgO_x$/Ag electrode-based devices demonstrate superior waterproofness compared to devices with $MoO_x$/Ag electrodes. These results demonstrate the wide applicability of the developed structure in enhancing waterproofness. Furthermore, devices based on PM6:ITCC active layer exhibit better PCE retention after immersion in water compared to other devices based on PM6:Y6, PBDTTT-OFT:IEICO-4F, and PM6:BTP-eC9. Optical images illustrate that the variations in waterproofness among different OPVs can be attributed to the water stability of the active layer (Supplementary Fig. 28). The irreversible damage of aggregation dots could be observed in the water soaked films of PM6:Y6, PBDTTT-OFT:IEICO-4F, and PM6:BTP-eC9, whereas the water soaked film of PM6:ITCC showed only the swelling streaks that disappeared after we removed the residual water. Additionally, we verified that using PEI-Zn as an ETL supports the improved water stability compared to the conventional sol-gel processed ZnO as ETL (Supplementary Fig. 29). Optical images illustrate that ZnO induces aggregation (Supplementary Fig. 30), likely due to the chemical dissolution of ZnO layer when exposed to water[54]. The evolution of $V_{OC}$, $J_{SC}$, FF, and PCE of double-side encapsulated PM6:Y6 OPVs under MPPT is shown in Supplementary Fig. 31. The double-side encapsulated PM6:Y6 OPV device exhibited operational stability with $T_{80}$ (80% of its initial value) lifetime of 27 h. Additionally, we measured the $T_{80}$ lifetime of devices based on PBDTTT-OFT:IEICO-4F, PM6:BTP-eC9, and PM6:ITCC active layers. Devices based on PBDTTT-OFT:IEICO-4F active layer demonstrated $T_{80}$ lifetime of approximately 110 h (Supplementary Fig. 32a), which is superior to the existing active layer solar cells (Supplementary Table 3). These results suggest that the differences in the operational stability of devices based on different active layers primarily stem from the active layer materials, and the operational stability of devices with $AgO_x$/Ag electrode could be further improved by developing a stable active layer. The evolution of $V_{OC}$, $J_{SC}$, and FF of double-side encapsulated solar cells based on PBDTTT-OFT:IEICO-4F under MPPT is depicted in Supplementary Fig. 32b. Notably, the decay of $V_{OC}$ in PM6:Y6 solar cells is

faster than that of $J_{SC}$ and FF, while the $V_{OC}$ of PBDTTT-OFE:IEICO-4F solar cell remains comparatively stable when compared to the variations in $J_{SC}$ and FF. The relatively swift decay of $V_{OC}$ in PM6:Y6 solar cells under operational stability could be ascribed to the changes in the microstructure of the active layer[55].

Furthermore, we investigated the waterproofness of OPVs with ternary active layers of PM6:Y6:$PC_{71}BM$, PBDTTT-OFT:IEICO-4F:$PC_{71}BM$, and PM6:BTP-eC9:$PC_{71}BM$ (Supplementary Fig. 1). The $J$–$V$ curves of ternary systems with in-situ grown $AgO_x$ HTL are shown in Supplementary Fig. 33. With the in-situ growth of $AgO_x$ HTL, these ternary OPVs demonstrated significant improvement in PCE, which is the same with the binary devices. The changes also suggest that the strategy with in-situ growth of $AgO_x$ HTL demonstrate excellent applicability for ternary solar cells with fullerene acceptor. All of these devices with ternary active layer systems also exhibited good waterproofness (Fig. 4f), further proving the universality of in-situ grown $AgO_x$ in enhancing the waterproofness. Moreover, these devices with ternary active layers demonstrated better operational stabilities compared to the devices with binary active layers (Supplementary Fig. 34). The best operational stability of free-standing OPVs was achieved using PBDTTT-OFT:IEICO-4F:$PC_{71}BM$ as an active layer, maintaining an estimated $T_{80}$ lifetime of 149 h. These results suggest that using $AgO_x$/Ag electrode strategy enables both waterproofness and operational stability by further optimizing the active layers.

## Discussion

In summary, we have successfully realized waterproof and ultraflexible OPVs with in-situ growth of $AgO_x$ HTL to strengthen the adhesion of the interface of HTL and the active layer. These OPVs offer remarkable stretchability and waterproof properties even with such thin structure, making them well-suited for wearable electronics. Notably, the efficiency degradation was limited to just 10% even after subjecting the devices to two washing cycles in a washing machine, each lasting 66 min. The optimization of waterproofness through structural design has enabled the OPVs to maintain excellent waterproof performance even with an ultrathin encapsulation layer. This approach has demonstrated broad applicability in enhancing the waterproof capabilities of ultraflexible OPVs. The successful implementation of waterproof organic OPVs holds promising implications for advancing self-powered wearable electronics and potentially underwater electronics for the Internet of Things in the foreseeable future.

## Methods

### Materials

PM6, Y6, BTP-eC9, IEICO-4F, and ITCC were purchased from 1-Materials (Quebec, Canada). PBDTTT-OFT was received from Toray Industries. $PC_{71}BM$ was purchased from Solenne BV Corporation. The precursor (ECRIOS VICT-Cz) of transparent polyimide was from Mitsui Chemicals (Japan). Zinc acetate dehydrates, 2-methoxyethanol, chloroform, and chlorobenzene were purchased from FUJIFILM Wako Pure Chemical Corporation (Japan). Ethoxylated polyethyleneimine (PEIE, 80% ethoxylated solution, 37 wt% in $H_2O$), 1-chloronaphthalene (1-CN), and 1,8-diiodooctane (DIO) were purchased from Sigma-Aldrich. All materials were used as received without further purification.

### Device fabrication

The tPI substrate with ITO electrode and PEI-Zn film were fabricated based on our previous study report[24]. PM6:Y6 (7.5 mg:9 mg) and PM6:Y6:$PC_{71}BM$ (7.5 mg:7.5 mg:1.5 mg) in chloroform with 1-chloronaphthalene (1-CN) solvent (1 mL, 99.5:0.5 volume ratio) was spin-coated at 3500 rpm for 40 s, followed by annealing at 110 °C for 10 min in an $N_2$-filled glovebox. PBDTTT-OFT:IEICO-4F (10 mg:15 mg) and PBDTTT-OFT:IEICO-4F:$PC_{71}BM$ (10 mg:12 mg:3 mg) in chlorobenzene with 1-CN solvent (1 mL, 97:3 volume ratio) was spin-coated

at 1400 rpm for 60 s, followed by annealing at 70 °C for 1 min in an $N_2$-filled glovebox. PM6:ITCC (10 mg:10 mg) in chlorobenzene with 1,8-diiodooctane (DIO) solvent (1 mL, 99.5:0.5 volume ratio) was spin-coated at 2000 rpm for 60 s followed by annealing at 100 °C for 10 min in an $N_2$-filled glovebox. PM6:BTP-eC9 (7 mg:8.4 mg) and PM6:BTP-eC9:PC$_{71}$BM (7 mg:7 mg:1.4 mg) in chlorobenzene with DIO solvent (1 mL, 99.5:0.5 volume ratio) was spin-coated at 2500 rpm for 60 s followed by annealing at 100 °C for 10 min. Ag (100 nm) or MoO$_x$/ Ag (7.5 nm/100 nm) were sequentially deposited (EX-200, ULVAC). The effective area of the solar cells was 4 mm². The devices with Ag electrode were annealed in air at 85 °C. The parylene (diX-SR, Daisan Kasei) encapsulation layer was evaporated by chemical vapor deposition (PDS 2010, KISCO Company). The removable electrode was fabricated by evaporating Cr and Ag on poly(dimethylsiloxane) (PDMS) substrate.

### Device characterization and measurement

The $J$–$V$ and stability measurements were performed under 1 sun illumination (XES-40S3, SAN-EI ELECTRIC) and recorded using a Keithley 2400 source in forward direction under 20 °C. The D-SIMS measurement (ADEPT-1010, ULVAC-PHI, Inc.) was performed in TORAY Research Center, Inc. The light intensity was calibrated by using a silicon reference diode (BS-520BK, Bunkoukeiki). The work function was measured using photoemission spectroscopy (RIKEN KEIKI AC-3). The ultraviolet (UV) photons emitted by a UV lamp of KEIKI were focused on the surfaces of the Ag electrode. The energy of UV light can be controlled by a spectrometer. The photoelectrons escaped from the Ag electrode surface because of photoelectric effect. The escaped photoelectrons were counted by an open counter. The work function is the energy of a point of interest between the baseline and extrapolated line in the figure of square root plots of the photoemission yield. The EQE measurements were performed using monochromatic light (SM-250F, Bunkoukeiki) calibrated using a silicon reference diode. The stretching–compressing test was conducted by laminating the free-standing OPVs on a pre-stretched elastomer (VHB Y-4905J, 3 M). A screw machine controlled by a program with pre-designed parameters was developed to control the strain of compression and stretching. The period for one compressing–stretching cycle is approximately 4 s. The adhesion measurement was performed by a high-precision tensile tester (AG-X, Shimadzu). The holder was gradually moved at 5 mm/min until the samples were completely separated from the stud. The contact angles were measured using a contact angle meter (DMe-211, Kyowa Interface Science). The dispersive energy ($\gamma_L^d$) and polar surface ($\gamma_L^p$) of deionized water are 21.8 mJ/m² and 51.0 mJ/m², respectively. The $\gamma_L^d$ and $\gamma_L^p$ values of glycerol are 37.0 mJ/m² and 26.4 mJ/m², respectively. Here, the surface energies of water and glycerol are average values[46]. Owing to the similar polarities between PM6:Y6 and AgO$_x$ ($\gamma^d > \gamma^p$), but contrasting polarities compared to PEDOT:PSS and MoO$_x$ ($\gamma^d < \gamma^p$), the calculation model for $W_a$ between AgO$_x$ and MoO$_x$ (or PEDOT:PSS) should be different. When calculating $W_a$ between PM6:Y6 and AgO$_x$, the $W_a$ can be calculated using the following equation[56]:

$$\mathrm{Wa} = 2(\gamma_1^d \gamma_2^d)^{1/2} + 2(\gamma_1^p \gamma_2^p)^{1/2} \qquad (1)$$

The $W_a$ between PM6:Y6 and MoO$_x$ (or PEDOT:PSS) should be calculated using the following equation[56]:

$$W_a = \frac{4\gamma^d \gamma_L^d}{\gamma^d + \gamma_L^d} + \frac{4\gamma^p \gamma_L^p}{\gamma^p + \gamma_L^p} \qquad (2)$$

The washing test was performed using a Panasonic washing machine (NA-FA80H7). The period of one standard washing cycle is 66 min, including washing, dewatering, and water injection procedures. The cross-sectional morphologies were observed by Thermo Fisher's SEM (QuattroS, Thermo Fisher Scientific) under 10 kV after Os

coat (2.5 nm). The cross sections were obtained through Ar ion milling for 12 h processing under 5.5 kV at 4 °C ± 5 °C (Leica, EM TIC 3X). The conductivity measurements were performed under dark and recorded using a Keithley 2400 source. The impedance tests were conducted using a potentiostat (AMETEK VersaSTAT 4) from 1 Hz to 500 kHz in the dark, at a bias voltage of 0.5 V.

### Reporting summary

Further information on research design is available in the Nature Portfolio Reporting Summary linked to this article.

## Data availability

The experiment data generated in this study are provided in the Source Data file. Source data are provided with this paper.

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

## Acknowledgements

We thank Dr. S. Lee and Dr. J. Kim of RIKEN and Dr. R. Wang of the University of Tokyo (Japan) for the fruitful discussions. This study was financially supported by the Japan Society for the Promotion of Science under its Grants-in-Aid for Scientific Research (No. JP22H04949), the Japan Science and Technology Agency (JST) under its Adaptable and Seamless Technology Transfer Program through Target-driven R&D (A-STEP) (No. AS3015021R).

## Author contributions

S.X., K.F., and T.S. conceived the idea and designed the research. S.X. fabricated and characterized the devices. S.L. fabricated the tPI substrates. K.F., K.N., and K.T. conceived the annealing strategy. S.X. and M.T. performed the compression test. Y.S. and M.T. conducted the adhesion measurement. D.I. and D.H. performed SEM characterization. B.D. performed the contact angle measurement. S. X., and Y. Z., designed and conducted the stability test. S.X., K.F., Y. Z., T.Y., and T.S. analyzed the data and designed the figures. S.X. wrote the first draft of the manuscript. K.F. and T.S. supervised the project. All authors have revised and approved the manuscript.

## Competing interests

The authors declare no competing interests.
