## [Peer Review File · Nature Communications]

Waterproof and ultraflexible organic photovoltaics with improved interface adhesionEditorial Note: This manuscript has been previously reviewed at another journal that is not operating a transparent peer review scheme. This document only contains reviewer comments and rebuttal letters for versions considered at *Nature Communications*.

REVIEWER COMMENTS

Reviewer #3 (Remarks to the Author):

After revisions, the work of Xiong and co-workers is now significantly more solid thanks to the greatly extended amount of data and discussions provided. Most importantly, the universality of their approach is satisfactorily demonstrated. In my opinion, the quality of the manuscript in its current form deserves publication in *Nature Communications*. Congratulations to the authors

Reviewer #4 (Remarks to the Author):

The manuscript reported waterproof and ultraflexible organic photovoltaics through the in-situ growth of a hole-transporting layer to strengthen interface adhesion between the active layer and anode. The author also added a ternary system to prove the universality of this strategy. Consequently, the 3- μm -thick OPVs demonstrated improved waterproofness compared to those using conventional HTL materials, retaining high stability of immersion and mechanical durability at strain underwater. However, I still have some doubts about this method. Besides, some data or analysis needs to be further supplemented to support the research results. Hence, this manuscript is not recommended for publication in this journal at the present stage, and the detailed comments are as follows:

1. In this paper, the author adds a ternary system to explore its universality, and the author should also consider the influence of water temperature on its stability.
2. The authors used three metal compounds (ZnO, AgO, and MoO₃) as the interface layer, respectively. Why does AgO have the best effect on strengthening interface adhesion between the active layer and anode?
3. Why annealing in air for 24 hours is the most conducive to in-situ growth of ZnO/HTL? The author should explain it in detail.
4. After the introduction of the double-sided encapsulation layer, how does water affect the interaction between metal oxides and the active layer, thereby affecting the stability of the device after being immersed in water?
5. Researchers generally choose vacuum evaporating MoO₃ as the interface layer, what are the advantages of choosing naturally oxidized metal materials as the interface layer compared with the

traditional way? And how to control the oxidation content of metals? The author should provide data representation as much as possible to supplement the description.

6. In this paper, it is said that Ag at the interface will be oxidized, so whether Ag on the surface of the device can be oxidized to AgO, will affect the conductivity of the electrode and ultimately affect the performance of the device.

7. It is recommended that the authors supplement conductivity and impedance tests to demonstrate the effect of metal oxides on stability.

Reviewer #5 (Remarks to the Author):

This is a very interesting paper on developing waterproof and ultra-flexible organic photovoltaics. The reviewer's comments are professional and strict, which greatly improved the quality of the manuscript. The authors have tried their best to provide detailed answers according to the reviewer's questions. Since the work is innovative and exhibits sufficient novelty, I recommend accepting and publishing the manuscript as soon as possible.

A small suggestion: the authors did not demonstrate much about the wearable properties or applications on their devices. I suggest removing the declaration of "wearable" in the title and main text to focus the study on waterproof capability, and it also prevents misleading the readers.

Reviewer #3:

Comment #3-0

After revisions, the work of Xiong and co-workers is now significantly more solid thanks to the greatly extended amount of data and discussions provided. Most importantly, the universality of their approach is satisfactorily demonstrated. In my opinion, the quality of the manuscript in its current form deserves publication in Nature Communications. Congratulations to the authors.

Reply #3-0

We appreciate Reviewer #3 for the positive evaluation of this work.

Reviewer #4:

Comment #4-0

The manuscript reported waterproof and ultraflexible organic photovoltaics through the in-situ growth of a hole-transporting layer to strengthen interface adhesion between the active layer and anode. The author also added a ternary system to prove the universality of this strategy. Consequently, the 3- μm -thick OPVs demonstrated improved waterproofness compared to those using conventional HTL materials, retaining high stability of immersion and mechanical durability at strain underwater. However, I still have some doubts about this method. Besides, some data or analysis needs to be further supplemented to support the research results. Hence, this manuscript is not recommended for publication in this journal at the present stage, and the detailed comments are as follows:

Reply #4-0

We appreciate Reviewer #4 for his/her time to evaluate our manuscript and for providing constructive comments to improve our manuscript. We revised our manuscript according to the comments.

Comment #4-1

1. In this paper, the author adds a ternary system to explore its universality, and the author should also consider the influence of water temperature on its stability.

Reply #4-1

We agree with this comment. We conducted additional waterproofness experiments to assess the stability under varying water temperatures. Specifically, we tested water temperature ranging from 0 °C (ice–water mixture) to 40 °C, which is the common temperature range in the environment. The devices developed with the structure of tPI/ITO/PEI-Zn/PM6:Y6/AgO_x/Ag/Parylene demonstrated excellent waterproofness under different water temperatures (0 °C, 20 °C, and 40 °C), suggesting that the water temperature ranging from 0 °C to 40 °C has no perceptible influence on the stability of devices. To clarify this point, the manuscript has been revised.

Our modification to the manuscript:

(Page 8, Lines 156–160)

“Additionally, we explored the influence of water temperature on the waterproofness. All devices based on AgO_x HTL demonstrated significant waterproofness under 0 °C, 20 °C, and 40 °C water immersion (Supplementary Fig. 14). The results indicate that the waterproof performance remains largely unaffected by water temperatures in the range of 0 °C–40 °C.”

(Supplementary Fig. 14)

Supplementary Figure 14 Waterproofness of OPVs under different water temperatures. The waterproofness tests of devices with structure $\text{rPI/ITO/PEI-Zn/PM6:Y6/AgO}_x/\text{Ag/Parylene}$ under different water temperatures.

Comment #4-2

2. The authors used three metal compounds (ZnO , AgO , and MoO_3) as the interface layer, respectively. Why does AgO have the best effect on strengthening interface adhesion between the active layer and anode?

Reply #4-2

We thank the reviewer for this comment. The interface adhesion strength between the active layer and anodes (AgO_x/Ag and MoO_x/Ag) can be quantitatively characterized by the thermodynamic work of adhesion (W_a), representing the necessary work to separate unit areas of the two phases in contact (Comyn, J. *Contact angles and adhesive bonding*. *Int. J. Adhes. Adhes.* 12.3 145-149 (1992)). As per calculation equations, W_a is related to the surface energies of these two in-contact materials. The surface energy of each materials is the sum of the dispersive (γ^d) and polar (γ^p) surface energies. Notably, the surface energy of MoO_x is much higher than that of AgO_x HTL. Furthermore, we noticed that the polarity of the active layer (PM6:Y6) is the same as that of AgO_x , whereas it is different from that of MoO_x . The different polarities between HTLs and active layer results in variations in the W_a calculation. The discussion of surface energy has been incorporated into the revised manuscript to clarify how the different polarities and surface energies of the HTLs (AgO_x and MoO_x) contribute to the divergence in W_a values, ultimately reflecting the strength of adhesion.

Our modification to the manuscript:

(Pages 12-13, Lines 261-278)

“The interface adhesion strength between the active layer and anodes (AgO_x/Ag and MoO_x/Ag) can be quantified using the thermodynamic work of adhesion (W_a). W_a represents the work needed to separate two in-contact materials⁴⁴ and is related to the surface energy (γ) of these materials. The γ of a film can be characterized by measuring the contact angles of liquid drops on the film⁴⁵. We measured the contact angles of deionized water and glycerol drops on films (Supplementary Fig. 25), and we utilized average surface energies of water and glycerol for accurate calculations⁴⁶. The dispersive surface energy (γ^d) and polar surface energy (γ^p) of AgO_x are 29.67 mJ/m^2 and 1.67 mJ/m^2 , respectively, resulting in a surface energy of 31.34 mJ/m^2 . For MoO_x , the γ^d and γ^p are 3.41 mJ/m^2 and 80.21 mJ/m^2 , respectively, yielding a

surface energy of 83.62 mJ/m^2 (**Supplementary Table 7**). Additionally, the different polarities of AgO_x ($\gamma^d > \gamma^p$) and MoO_x ($\gamma^d < \gamma^p$) indicate that the calculation of W_a should be different (See Methods section for detailed information). These differences in polarities and surface energies result in varying W_a values between the PM6:Y6 active layer and AgO_x/Ag (44.08 mJ/m^2) and MoO_x/Ag (24.55 mJ/m^2) anodes. A higher W_a between the active layer and anode indicates a stronger interface adhesion. Hence, the replacement of the anode from MoO_x/Ag to AgO_x/Ag significantly improves the interface adhesion.”

(References)

- [44] Packham DE. Work of adhesion: contact angles and contact mechanics. *Int. J. Adhes.* 16, 121-128 (1996).
- [45] Hntsberger JR. Surface energy, wetting and adhesion. *J. Adhes.* 12, 3-12 (1981).
- [46] Fowkes FM. Attractive forces at interfaces. *Ind. Eng. Chem.* 56, 40-52 (1964).

Comment #4-3

3. Why annealing in air for 24 hours is the most conducive to in-situ growth of ZnO/HTL? The author should explain it in detail.

Reply #4-3

We appreciate the reviewer for this comment. The thermal annealing process in air plays a crucial role in the in-situ growth of the energy-level-matched AgO_x HTL. It is essential to determine the appropriate annealing time for proper electrode oxidation. Insufficient oxidized time hinders the proper formation of the AgO_x HTL at the interface, resulting in poor charge-carrier extraction efficiency. Conversely, excessive annealing time leads to the electrodes, interfaces, and active-layer deterioration, affecting the performance of the devices. We have incorporated these clarifications into the revised manuscript.

Our modification to the manuscript:

(Page 6, Lines 114-123)

“The devices annealed for 24 h at $85 \text{ }^\circ\text{C}$ demonstrated the best average performance (**Supplementary Table 4**), highlighting the crucial role of the appropriate annealing in the formation of an efficient AgO_x HTL. **Supplementary Fig. 8** illustrates the evolutions of different annealing times of photocurrent density (J_{ph})–effective voltage (V_{eff}) curves and light intensity-dependent J_{sc} . These results suggest that devices annealed for 24 h at $85 \text{ }^\circ\text{C}$ experienced a significant improvement in the collection and extraction of charge carriers, as well as the suppression of bimolecular recombination. However, the excessive annealing can cause performance deterioration^{24, 31}. Consequently, the prolonged heating could not yield further enhancements in device performance.”

Supplementary Figure 8 Analysis of extraction and transportation of charge carriers. a, The curves of J_{ph} versus V_{eff} in the devices (tPI/ITO/PEI-Zn/PM6:Y6/AgO_x/Ag/Parylene) for different annealing times at 85 °C. b, The dependence of P_{light} on J_{SC} of the devices (tPI/ITO/PEI-Zn/PM6:Y6/AgO_x/Ag/Parylene) under different annealing times at 85 °C.

Supplementary Table 4. Average photovoltaic parameters of ultraflexible OPVs annealing at 85 °C in air from 12 h to 36 h.

Annealing time (h)	J_{SC} (mA cm ⁻²)	V_{OC} (V)	FF	PCE (%)
12	24.0 ± 1.2	0.76 ± 0.01	0.70 ± 0.03	12.8 ± 0.3
16	25.1 ± 0.2	0.76 ± 0.01	0.70 ± 0.01	13.4 ± 0.2
20	24.9 ± 0.3	0.77 ± 0.01	0.71 ± 0.01	13.7 ± 0.2
24	25.4 ± 0.4	0.78 ± 0.01	0.70 ± 0.01	13.9 ± 0.3
28	24.6 ± 0.5	0.79 ± 0.01	0.71 ± 0.01	13.8 ± 0.5
32	24.7 ± 0.2	0.79 ± 0.01	0.70 ± 0.01	13.7 ± 0.3
36	24.7 ± 0.3	0.79 ± 0.01	0.70 ± 0.01	13.7 ± 0.3

These are statistical values of average and standard deviation obtained from 12 samples.

(References)

- [24] Xiong S, et al. Ultrathin and efficient organic photovoltaics with enhanced air stability by suppression of zinc element diffusion. *Adv. Sci.* 9, 2105288 (2022).
- [31] Yang W, et al. Simultaneous enhanced efficiency and thermal stability in organic solar cells from a polymer acceptor additive. *Nat. Commun.* 11, 1218 (2020).

Comment #4-4

4. After the introduction of the double-sided encapsulation layer, how does water affect the interaction between metal oxides and the active layer, thereby affecting the stability of the device after being immersed in water?

Reply #4-4

We thank the reviewer for this comment. The water vapor transmission rate (WVTR) of the parylene encapsulation layer is approximately 90 g·m⁻²·day⁻¹. While the introduction of the double-side encapsulation layer helps alleviate the impact of water on the devices, it cannot provide complete prevention. Previous studies suggest that the WVTR for OPVs barriers should be lower than 10⁻³ g·m⁻²·day⁻¹. Therefore, the protection afforded by double-side encapsulation layers could not entirely eliminate the influence of water. We have clarified this point in the revised manuscript.

Our modification to the manuscript:

(Page 14, Lines 294-296)

“The encapsulation layer of parylene has a water vapor transmission rate (WVTR) of approximately 90 g/m² per day¹⁸, inadequate for fully preventing the impact of water on the devices⁵³.”

(References)

[18] Xu X, et al. Thermally stable, highly efficient, ultraflexible organic photovoltaics. Proc. Natl. Acad. Sci. USA 115, 4589-4594 (2018).

[53] Cros S, et al. Definition of encapsulation barrier requirements: A method applied to organic solar cells. Sol. Energy Mater. Sol. Cells 95, S65-S69 (2011).

Comment #4-5

5. Researchers generally choose vacuum evaporating MoO₃ as the interface layer, what are the advantages of choosing naturally oxidized metal materials as the interface layer compared with the traditional way? And how to control the oxidation content of metals? The author should provide data representation as much as possible to supplement the description.

Reply #4-5

We appreciate the reviewer for this comment. While vacuum evaporating MoO₃ is widely used as the hole transporting layer (HTL), it is also sensitive to water or moisture. In our study, we primary focus is on the waterproofness of the devices. The device based on MoO₃ HTL demonstrated poor waterproofness, whereas the device based on oxidized AgO_x HTL demonstrated excellent waterproofness even under deformation, making them suitable for achieving the ultraflexible and waterproof organic solar cells. The comparison of waterproofness between devices based on MoO_x and AgO_x HTL has been illustrated in the original version (Fig. 2 and Supplementary Fig 20).

Quantitatively determining the oxidation content of the metal is difficult. The formed AgO_x layer does not show a sharp interface with the Ag electrode and active layer after the thermal annealing-assisted oxidation process. Content analysis of Ag and O elements by SIMS for AgO_x HTL could be influenced by Ag from the electrode and O from the active layer, leading to incorrect calculations. The optimal oxidation condition is determined by changing the annealing time.

Comment #4-6

6. In this paper, it is said that Ag at the interface will be oxidized, so whether Ag on the surface of the device can be oxidized to AgO, will affect the conductivity of the electrode and ultimately affect the performance of the device.

Reply #4-6

We thank the reviewer for this helpful comment. We conducted additional measurements of the work function of the top surface of the Ag electrode. The work function changes from 4.67 eV to 4.69 eV after 24 h annealing. Meanwhile, we measured the conductivity of the electrode and observed no significant deterioration. These results suggest that the 24 h annealing does not induce severe oxidation to the Ag on the surface. The observation could be attributed to the fact that oxygen diffusion across grain boundaries is much faster than lattice diffusion. The silver electrode generally grows in the Volmer–Weber mode (‘island’ mode) on the active layer, resulting in a rougher surface

and a higher grain boundary density in the Ag–organic interface than at the top surface of Ag electrode. The rougher surface and a higher grain boundary indicate quicker oxygen permeation in the Ag-organic interface than at the top surface of the Ag electrode. A more detailed discussion on these points has been included in the manuscript for clarification.

Our modification to the manuscript:

(Pages 7-8, Lines 138-152)

“Subsequently, we checked the work function of the Ag electrode before and after annealing. Notably, the work function of top surface of the annealed Ag electrode is almost the same as that of a fresh Ag electrode, and the conductivity remains unchanged (**Supplementary Fig. 12**). These indicate that the 24 h annealing does not cause significant oxidation on the Ag surface, thereby not affecting the performance. Here, we applied a removable electrode³⁴ to assess the work function of the bottom surface of the Ag electrode in contact with the active layer (**Supplementary Fig. 13a**). The annealed removable Ag electrode exhibited a work function of approximately 5.0 eV (**Supplementary Fig. 13b**), similar to the reported value³⁵. Notably, silver oxidizes faster in the Ag-organic interface during annealing compared with the top surface of the Ag electrode because oxygen diffusion across grain boundaries is quicker than lattice diffusion³⁶. The silver electrode generally grows in the Volmer–Weber mode (‘island’ mode) on an active layer³⁷, leading to a rougher surface and higher grain-boundary density in the Ag–organic interface than at the top surface of Ag electrode³⁸. The energy-level diagram of the optimized devices is shown in **Fig. 1e**.”

(Supplementary Fig 12)

Supplementary Figure 12 Work function and conductivity of the electrode. a, Work function of the top surface of the Ag electrode before annealing. b, Work function of the top surface of the Ag electrode after 24 h annealing at 85 °C in air. c, Demonstration of the conductivity measurement for the common Ag electrode. Scale bar is 5 mm. d, Conductivity of the Ag electrode before and after

annealing.

(References)

- [34] Shimano S, Fukuda K, Someya T, Yokota T. Development of air-stable photomultiplication-type organic photodetector and analysis of active layer using removable top electrode. *Adv. Electron. Mater.* 8, 2200651 (2022).
- [35] Das S, Alford TL. Improved efficiency of P3HT:PCBM solar cells by incorporation of silver oxide interfacial layer. *J. Appl. Phys.* 116, 044905 (2014).
- [36] Hoffman RE, Turnbull D. Lattice and grain boundary self-diffusion in silver. *J. Appl. Phys.* 22, 634-639 (1951).
- [37] Sennett RS, Scott GD. The structure of evaporated metal films and their optical properties. *J. Opt. Soc. Am.* 40, 203-211 (1950).
- [38] Kim JB, Kim CS, Kim YS, Loo Y-L. Oxidation of silver electrodes induces transition from conventional to inverted photovoltaic characteristics in polymer solar cells. *Appl. Phys. Lett.* 95, 183301 (2009).

Comment #4-7

7. It is recommended that the authors supplement conductivity and impedance tests to demonstrate the effect of metal oxides on stability.

Reply #4-7

We agree with this comment. We have incorporated supplementary conductivity and impedance tests into the revised manuscript. The conductivities of both AgO_x/Ag and MoO_x/Ag electrode exhibited excellent stability after water immersion. This implies that the electrode conductivity does not impact the stability of the devices after immersion. Notably, the transport resistance (intersecting point of the curve and the lateral axis at low frequencies) of the device based on AgO_x/Ag electrode slightly increased after water immersion, whereas the transport resistance in the MoO_x/Ag -based device increased sharply. The drastically elevated transport resistance in the device-based MoO_x/Ag electrode indicates the generation of more defects and charge recombination only after 1 h water immersion. Because both devices have the same structure except for the HTL layer, the difference definitely resulted from the charge transportation at the HTL layer.

Our modification to the manuscript:

(Page 8-9, Lines 163-175)

*“Then, we examined the electrode conductivity of the waterproof devices before and after water immersion. Both AgO_x/Ag and MoO_x/Ag electrodes maintained a stable conductivity after water immersion (**Supplementary Fig. 16a and 16b**), implying that the electrode conductivity does not impact the water stability of the devices. To delve deeper, impedance tests were conducted, and the impedance spectra of the device before and after water immersion are shown in **Supplementary Fig. 16c and 16d**. The transport resistance³⁹ (intersecting point of the curve and the lateral axis at low frequencies) of the device based on AgO_x/Ag slightly increased after water immersion, whereas that of the device with MoO_x/Ag sharply increased only after 1 h immersion. The increased resistance indicates an increase in charge recombination after water immersion^{39, 40, 41}. Because both devices share the same structure except for the HTL layer, the observed difference can be attributed to the charge transportation*

within the HTL layer.”

(Page 21, Lines 443-446)

“The conductivity measurements were performed under dark and recorded using a Keithley 2400 source. The impedance tests were conducted using a potentiostat (AMETEK VersaSTAT 4) from 1 Hz to 500 kHz in the dark, at a bias voltage of 0.5 V.”

(Supplementary Fig 16)

Supplementary Figure 16 Conductivity of electrodes and the impedance spectra of devices. a, Conductivity of AgO_x/Ag electrode before and after water immersion. The inset is the demonstration of the conductivity measurement of the common electrode. Scale bar is 1 cm. b, Conductivity of the MoO_x/Ag electrode before and after water immersion c, Impedance spectra of devices based on the AgO_x/Ag electrode before and after water immersion. d. Impedance spectra of devices based on the MoO_x/Ag electrode before and after water immersion.

(References)

- [39] Jing J, et al. Semitransparent organic solar cells with efficiency surpassing 15%. *Adv. Energy Mater.* 12, 2200453 (2022).
- [40] Liu L, et al. Nanographene–osmapentalene complexes as a cathode interlayer in organic solar cells enhance efficiency over 18%. *Adv. Mater.* 33, 2101279 (2021).
- [41] Pockett A, et al. A combined transient photovoltage and impedance spectroscopy approach for a comprehensive study of interlayer degradation in non-fullerene acceptor organic solar cells. *Nanoscale* 11, 10872-10883 (2019).

Reviewer #5:

Comment #5-0

This is a very interesting paper on developing waterproof and ultra-flexible organic photovoltaics. The reviewer's comments are professional and strict, which greatly improved the quality of the manuscript. The authors have tried their best to provide detailed answers according to the reviewer's questions. Since the work is innovative and exhibits sufficient novelty, I recommend accepting and publishing the manuscript as soon as possible.

Reply #5-0

We thank the reviewer for the positive comments on this work.

Comment #5-1

A small suggestion: the authors did not demonstrate much about the wearable properties or applications on their devices. I suggest removing the declaration of "wearable" in the title and main text to focus the study on waterproof capability, and it also prevents misleading the readers.

Reply #5-1

We agree with the reviewer's suggestion. We removed the declaration of the wearable from the title and main text.

Our modification to the title:

"Waterproof and ultraflexible organic photovoltaics with improved interface adhesion"

REVIEWERS' COMMENTS

Reviewer #4 (Remarks to the Author):

In the manuscript "Waterproof and ultra-flexible organic photovoltaics with improved interface adhesion", Xiong et al nicely reported an idea to produce ultra-flexible organic photovoltaics through the in-situ growth of a hole-transporting layer to strengthen interface adhesion between the active layer and anode. Compared with conventional sequentially-deposited hole-transporting layers, the in-situ grown hole-transporting layer exhibits higher thermodynamic adhesion between the active layers, resulting in better waterproofness. This strategy provides valuable insights into the fabrication of ultra-flexible organic photovoltaics. The authors have fully responded to the reviewers' comments and the article has been revised in detail. The work exhibits sufficient innovation, so I recommend accepting the manuscript in Nature Communications.